# Speak Catalan to Me, I'm a Catalan Muslim Woman: Producing Proposals for Religious and Education Policy through Participatory Research from a Gender Perspective

Assumpta Aneas *, Núria Lorenzo Ramírez, Marta Simó Sánchez and Alba Ambrós Pallarés

Faculty of Education, University of Barcelona, 08007 Barcelona, Spain; nuria.lorenzo@ub.edu (N.L.R.);
martasimo@ub.edu (M.S.S.); aambros@ub.edu (A.A.P.)
* Correspondence: aaneas@ub.edu

**Abstract:** Specific groups of Catalan citizens, in spite of them being socially and professionally integrated, suffer the risk of exclusion or segregation on grounds of identity, one example being those who identify with Islam. This study arises from a prospective research project centred on a case study with the Catalan Muslim Women's Association. The main objective was to formulate public policy proposals on education, religion, and gender to be included in the Citizenship and Immigration Plan, through a process based on the women's participation and testimony. The study was divided into two phases: the participatory research followed by evaluation of the applicability of the resulting proposals. It was conducted through semi-structured interviews ($n = 37$), a discussion group ($n = 21$), and a round table ($n = 31$). Oral and textual qualitative data were gathered and analysed using the Ivàlua logical framework. Results for education policy urge the creation of a new professional specialist mediating between all actors. Those for religion call for public spaces for religious practice. In relation to gender, participants continued to demand policies that do not discriminate against Muslim women. In summary, religion is a resource that supports identities, beliefs, and practices, with both empowering and disempowering effects.

**Keywords:** gender; religiosity; education; Responsible Research and Innovation (RRI); public polices; women



## 1. Introduction

### 1.1. Context of the Study

This article presents the proposals arising from a participatory, prospective study whose aim was to provide policy guidelines for the Catalan government, in response to the question of how to encourage the feeling of belonging in a multicultural community, through which to produce relevant knowledge for decision making on citizenship and immigration (Aneas 2019; Aneas et al. 2020). The study was financed by the Research Area of the Catalan School of Public Administration (EAPC) through a public tender awarded to the Intercultural Education Research Group GREDI at the University of Barcelona. The main requirement of the tender was that the Responsible Research and Innovation (RRI) approach should be applied to a participatory study. In this approach, non-academic social actors (representatives of civil society and the education community, and other public decision makers) participate in and co-lead together with the academic research team, as Rip (2018) explains. The tender also stipulated that the study results should produce specific policy proposals to form part of the government's 2021–2024 Citizenship and Immigration Plan. These principles are fully in accordance with the guidelines of the eighth Framework Program, Horizon Europe (2021–2027), which directs research towards social action; that is, towards the solution of real social challenges and problems. The project also fulfilled European Commission policies, which encourage socially relevant, transdisciplinary research involving a range of stakeholders (academia, public decision

makers, industry, civil society, and the education community). A further requirement was to provide multiple bottom–up solutions, i.e., from the bottom to the top of the political ladder. This article describes the most important parts of the project with a two-fold objective: first, to enable the experience to be replicated in other contexts, and second, to present the proposals decided on for the fields of education, religion, and gender.

Catalonia has a population of 8,005,784 (2023) and is the region with the second highest number of inhabitants in Spain, with 17% of the country's total (INE 2023). Women make up the majority at 51% (Idescat 2023). Catalonia is home to 641,101 Muslims, the largest Islamic population in Spain. At the end of 2022, numbers of Spanish-nationality Muslims (250,000) overtook those of Moroccan nationality (230,000), with Pakistani Muslims in third place at almost 56,000 (Fernández 2023).

The initial spur for this project came from the 2017 terrorist attacks in Catalonia.[1] The discovery that the perpetrators had been young Muslims of Moroccan origin who had grown up and attended school in their cities of residence aroused deep consternation. In addition, it raised doubts and questions around the issue of how to achieve effective integration of religious minorities in Catalonia. In response to this situation, Muslim women who are also Catalan citizens were given a leading role in this study, to bring their perceptions and feelings to bear in formulating public policy proposals with a gender perspective. This was, in short, a project in which the two key variables of gender and religion were intertwined in a society that aims to be inclusive and egalitarian.

The premises and theoretical assumptions underlying the project are presented below.

### 1.2. Policy and Dialogue in the Area of Religious Plurality

As Lehmann (2020) remarks, the concepts of religious plurality and pluralization are central themes in the academic debate on interreligious dialogue (IRD). He also notes that for some decades now researchers have advocated for dialogue as a means of addressing the challenges of the coexistence of diverse religious traditions in specific socio-cultural contexts. Recently, the IRD movement has focused its attention on relationships and encounters between multiple religious actors, both individually and collectively and in both public and private arenas. These encounters have revealed a range of diverse needs, political agencies, and frictions between differing legal frameworks (Iwuchukwu 2013). In the case of Spain, the state is secular and non-confessional, with freedom of worship and the right to receive religious education enshrined in law (Gobierno de España 1978 Article 27.3 of the Spanish Constitution).

Griera (2016) summarizes the development of public policy on religious matters in Catalonia. In this context, it is essential to remember that legislative powers in the area of migration pertain exclusively to the Spanish central government as opposed to the regional Catalan administration. We recall this point because the original major driving force in the growth of religious diversity in Spain was the arrival of non-EU immigrants from the 1990s onwards. Currently, there are thousands of practitioners of the Muslim, Protestant, Hindu, etc., faiths who have Spanish nationality. It is important to remember these beginnings to understand why so much attention has been given to specifically religious issues in Catalonia, since in this area, unlike that of immigration, the regional government does have regulatory power. Thus, in 2000, the Secretariat for Relations with Religious Confessions (Secretariat de Relacions amb les Confessions Religioses, currently known as the General Directorate of Religious Affairs) was created, with the aim of taking a more active role in this policy sector. This was a pioneering initiative in Spain, as no other regional government had developed an explicit policy program in the area, and even central government policy on religious affairs had a low public profile at the time. In 2000, then, the Catalan government put in place an ambitious policy agenda on religious issues, whose key lines have been ratified with a broad consensus in the Catalan Parliament. Four major political objectives frame this approach to religious plurality, summed up by Griera as follows:

The first objective was to "promote a social agreement for secularism and religious pluralism in Catalan society" through signing agreements with religious minorities and pursuing a policy of inclusive reconciliation at the local level and within public institutions (schools, hospitals, and prisons). The second objective was for the regional government to extend its control over religious affairs by claiming more legal, economic, and political jurisdiction in the issue. The third objective was "to foster the Catalanisation of religious hierarchies" by "avoiding interference from foreign countries in this issue" and by encouraging religious leaders to learn the Catalan language and acquire knowledge of Catalan culture; thus, in this area, the institutionalisation of representative religious bodies was stressed in order to foster dialogue between the government and the different faiths. The fourth objective was to promote interreligious dialogue and good relations between religious communities, political actors, and civil society organizations, and to set up the necessary political bodies to put this into effect (Griera 2016, p. 11).

Religious practice also poses issues for a broad definition of citizenship, and norms and practices are often contrary to secular laws on gender equality and international human rights conventions. In a sense, religious institutions that maintain strong connections in both public and private spheres can be seen as "borders" where the negotiation of the exercise of citizenship can be especially complex.

### 1.3. Women and Religion

Religious beliefs, practices, and organizations are themselves gendered, insofar as women and men are assigned different rights and duties and exercise different religious roles within them. Certainly, women must work, adjusting at the patriarchal characteristics of their own religion, but as active agents they are able to find ways to gain power and influence through religious faith and practice. It is necessary to take into consideration the active role, organizing and managing activities and resources, played by women in religious organizations (Warner 1993). This is of prime importance when we remember that the religious sphere is one of the social sites in which gender relations are formed and negotiated (Predelli 2008) in contemporary societies.

Focusing directly on the role of women in Islam, it should be remembered that the coming of Islam involved a substantial improvement in women's rights in seventh-century society, characterized by patriarchy and androcentrism (Llorent Bedmar 2013; Seedat 2013). In the West, however, Islam is often seen as a homogeneous religion that is incompatible with gender equity, generally ignoring the enormous cultural, political, and economic differences amongst its followers (Blasco 2018). However, as Alhareth et al. (2015) point out, many of the practices that undermine gender equality in Islamic countries have their origin in the cultural traditions and policies of the specific country rather than Islam itself. On the other hand, it should also be stressed that Islam envisages a division of roles and responsibilities between men and women in terms of what it sees as their innate nature (Qur'an 92:3-4), considering the two roles to be complementary (Seedat 2013). This has generated substantial debate around the question of whether Islam promotes gender inequality (Bauer 2009).

### 1.4. Minoritization, a Way to Inclusion as Citizen, to Exclusion, or to Radicalization

It should be remembered that European societies, regarding culture, are no longer homogeneous, with a White population and the Judeo-Christian tradition as the norm. Today, the cultural, religious, and social fabric has become much more heterogeneous and complex. Yet, due to the still-prevailing image of homogeneity, people and groups who do not fit into this hegemonic pattern are considered subaltern minorities, outsiders, or simply invisible (Barker 1981; Giménez Romero 2003). This has important consequences in terms of social and collective identity and power. Thus, in our view, it is necessary both to reveal and display the richness of our contemporary social melting pot. This is, or should be, a society in which all citizens, regardless of their origins and beliefs, can feel belonging and exercise their citizenship fully (Hobsbawm 2012; Anderson 1991; Castells 1998; Habermas

1998; Bauman 2003). Thus, we must ensure that these citizens are known, recognized, and valued, and that their legacies and traditions are acknowledged by all as a form of wealth that enriches society as a whole (Holz 2001; Giesen 1999).

The forms of minoritization remarked on above affect not only cultural factors such as religion but are also gender-related. A majority religion usually perceives itself as objective, universal, and non-negotiable truth in comparison to minority religions, whose "strange" and "incomprehensible" practices should be changed or forbidden. In this study, the concept of a religious minority refers to numerical minorities. The word "minoritized" denotes the relationship between a group in a position of majority and power to other, less privileged groups. The concept of minoritization involves the attribution to such groups of thoughts and practices that are associated with inferiority, backwardness, and/or oppression. Thus, in the religious context, and in its gender structures, relations between the majority and the minorities are affected by power (Roald 2005, p. 20). Thus, women (a minoritized group) who are also Muslims face additional barriers and limitations, simply because their religious practices and traditions are also minoritized, and thus, they are a "minority within a minority" (Eisenberg and Spinner-Halev 2005), suffering double minoritization.

Analyses of radicalization in social psychology suggest that it may be a response to identity-based exclusion and/or existential frustration (Gephart 1999; Piqueras Infante 1996; May 1968; Frankl 2010). If we see integration as the "process of being accepted as part of society" (Penninx et al. 2006, p. 124), in its cultural and religious dimensions, then there are some people of Spanish nationality, who may even have Catalan surnames and be of medium or high economic status, who suffer stereotyping, minoritization, and, sometimes, discrimination. In these cases, this can cause a feeling of being rejected and a lack of belonging, which in turn can create a feeling of injustice and exclusion, ultimately triggering radicalization (Gurr [1970] 2012). Lastly, we should note, as remarked above, that in addition to exclusion, radicalization is an enormous risk, the great fear towards whose prevention all political and educational efforts are directed. Although it is not the aim of this article to address this issue, since, as Bourekba (2018) remarks, the nature of radicalizing factors is an ongoing debate, in our view, it is important to address it through a multidisciplinary approach.

It is important to focus on addressing the needs and perspectives of Muslim women as a group, not only because of their rights as citizens, but also due to their crucial role in the family structure and in the socialization and acculturation process (Nava et al. 2014). In the same area, Navarro-Granados (2022) has studied the issue of Muslim female radicalization, and summarizes the factors that seem to promote such radicalization and, in some cases, the recruitment of women by fundamentalist organizations. The reasons for Muslim women deciding to join Islamist-inspired groups are found in the pull and push factors listed in Table 1, identified in studies by the Radicalization Awareness Network Centre of Excellence (RAN Centre of Excellence) and the International Centre for the Study of Radicalization (ICSR; also cited in Navarro-Granados 2022). We present these findings here because, as is shown in the empirical section of this study, some of these same factors underlie the feelings of the women of the Associació de Dones Musulmanes a Catalunya (Catalan Muslim Women's Association, ADMAC).

The theoretical framework presented so far informed the objectives of the study, namely, (1) to document the Catalan Muslim women's experience of their social, family, and spiritual life; (2) to analyse and synthesize dialogically the challenges and responses emerging from the experiential views of the women of the Catalan Muslim Women's Association (ADMAC), from the perspective of services and policies; and (3) to formulate specific policy proposals in the fields of education, religion, and gender.

**Table 1.** Motivations for Muslim women deciding to join violent Islamic-inspired groups.

|  | Factors | Indicators |
|---|---|---|
| **Pull factors** | Feeling isolated in Western culture | - Ethnic and religious discrimination<br>- Identity conflict (adolescence) |
|  | Feeling that the Muslim community is persecuted worldwide | - Feeling of oppression towards the Ummah<br>- Narrative of "us against them"<br>- Belief in "a war on Islam" |
|  | Frustration at inaction from the international community | - Justification of violence against "enemies"<br>- Blaming the West for the sufferings of the Ummah<br>- Fighting for a "just cause" |
| **Push factors** | Religious duty to build an Islamic state | - Religious duty to carry out the hijra<br>- Rewards in paradise for this decision<br>- Desire to build a society governed by a strict interpretation of the Shari'ah |
|  | Sense of community, fraternity, and belonging | - Feeling that they belong to an Umma with which they share goals and lifestyle<br>- Participating and acting for a common cause |
|  | Romanticism and adventure | - Desire to marry a mujahideen |

Sources: RAN Centre of Excellence (2015); Saltman and Smith (2015) in ICSR Report; Navarro-Granados (2022).

## 2. Results

This section presents only the outcomes of the assessment of the applicability of the ADMAC proposals as public policies to be incorporated in the Citizenship and Immigration Plan. To create the following tables, as explained above, a series of steps were undertaken, synthesizing the information, analysing it in terms of importance and relevance, and drafting it in the form of policy proposals. The proposals cover the three topic areas of education, religion, and gender. They are presented here according to Ivàlua's logical framework, divided into three areas: issues, activities, and expected changes.

In the area of religion (Table 2), factors emerged relating to integration and acceptance not only in the host society but also in the Muslim minority itself. The complexity of the feeling of identity, which in many cases may be multiple and which should also be accepted and understood by all, was evidenced. In addition, participants particularly stressed structural factors regarding the lack of places and means for fully exercising religious practice, since such spaces would facilitate coexistence, acceptance, and incorporation into society's everyday life. Also, putting into practice hybrid models of cultural practice would facilitate transparency and openness to all those involved. Time off work was also called for, in order to celebrate Muslim events such as Friday prayers or Ramadan. These changes would enhance the feeling of belonging to society and thus strengthen the development of non-threatened identity.

**Table 2.** Public policy proposals for the religious sphere.

| **Public Policy Proposals for the Religious Sphere** |
| --- |
| Issues |
| - Difficulty in practicing religion in all spheres.<br>- Conflicts of loyalty between various cultural and religious traditions that prevent the development of diverse identities in harmony, from the standpoint of complementarity rather than incompatibility.<br>- A very urgent problem was the case of young people with multiple, antagonistic identity constructions.<br>- Mistrust (caused by ignorance) of what happens in mosques. |
| Activities |
| - Creating a space for religious practice in public places, like a breastfeeding room, so that it is visible and recognized.<br>- Enabling workers to take time off to attend religious festivities. Young people ask for places where they can speak and express themselves without being judged, and to have religious texts translated into a language they understand, i.e., Catalan and Spanish, not only Arabic. |
| Expected changes |
| - Structural changes: the opportunity for everyone to participate in religious events in public areas.<br>- Hybrid models of cultural practice in which diverse sources of identity and traditions can coexist, consistent with the real diversity of identities. For example, a Ramadan calendar as well as an Advent calendar. How to celebrate the Christian Christmas, Kings' night, in mixed families, etc. |

As the three areas in Table 3 show, it is essential to work in the cultural and religious sphere to combat the feeling of exclusion. Thus, the educational dimension is vital, as it is still one of the main socializing institutions and therefore paramount in the fight against radicalization. In order to ensure dialogue on issues of religious diversity within the educational sphere, participants particularly stressed the creation of a new professional specialist trained in interreligious dialogue who would liaise between families, schools, students, teachers, and the administration. In this way, safer and more comfortable spaces could be built both inside and outside schools. At the same time, the visibility of this new specialist would help to normalize the situation for all citizens by creating a feeling of belonging and respect towards the Muslim community, without resorting to surveillance measures, which usually create a feeling of alienation (Bourekba 2018).

**Table 3.** Public policy proposals for education.

| **Public Policy Proposals for Education** |
| --- |
| Issues |
| - Difficulty in experiencing one's religious identity naturally.<br>- Difficulty in acceptance of difference and being accepted if one is different or belongs to a minority.<br>- Becoming visible, being recognized, and participating in family diversity at school.<br>- Insecurities around and fears of addressing these issues in class and at school; concealing oneself.<br>- Managing the diversity of practices, beliefs, and knowledge making up the educational ecosystem. |

**Table 3.** *Cont.*

| Public Policy Proposals for Education |
| :---: |
| Activities |

- Creating a new professional profile in Intercultural Competence and Interreligious Dialogue as part of a competency-based teaching degree.
- A new professional post for helping and counselling families and enhancing their relationship with the educational ecosystem.
- Developing binding regulations to protect and ensure certain rights and duties of minoritized groups.
- Stipulating the development of intercultural competencies and interreligious dialogue in the short, medium, and long term at all levels of the educational system.
- Stipulating development aimed at positive management of differences in beliefs and conflict.
- Recognizing, training, and empowering all families and involving them in the educational institutions at all stages, through a range of strategies in the short, medium, and long term.
- Creating new spaces and models.
- Disseminating and taking advantage of the many existing resources.
- Applying passion, realism, and innovation to prevent frustration.

| Expected changes |
| :---: |

- Structural changes: welcoming, organizational regulations, curriculum content.
- A plural, cohesive educational community in which people feel they have the right to be different.
- Changes in behaviour and attitudes (citizens with intercultural and interreligious competence).

Finally, around gender, the results (Table 4) emphasized the need to continue developing policies that do not discriminate against Muslim women for wearing the hijab and that respect them for this in all spheres of society. Although the good work that the Spanish state has already achieved was highlighted, the creation of a regulatory framework guaranteeing the right to one's own gender identity linked to religion was called for. Protection of Muslim women suffering gender violence, as also unfortunately occurs with non-Muslim women, was also demanded.

**Table 4.** Proposals for public policy in the area of gender.

| Proposals for Public Policy on Gender |
| :---: |
| Issues |

- Difficulty in experiencing religious identity naturally.
- Misogyny, Islamophobia, and Western feminism, which also attacks certain values and practices.
- Abuse of the hijab to divert attention from other issues, individualizing the victim and leaving responsibility and choice up to the individual, while not respecting basic citizens' rights.
- This is NOT a religious issue but a gender issue. There is no problem with the patka, the topknot worn by male Sikh children, for example.

| Activities |
| :---: |

- Develop binding regulations to ensure and protect the rights and duties of minoritized groups.
- Develop competency training for social workers and other relevant professionals.
- The public administration should be a role model, developing good practices and becoming a base for awareness raising.

**Table 4.** *Cont.*

| Proposals for Public Policy on Gender |
| --- |
| Expected changes |

- Structural changes: a regulatory framework guaranteeing the right to one's own religious identity.
- That there should be NO discrimination against women wearing the hijab in access to jobs, education, and public services.
- New facilities for and models of community relations for women, youth, and adults, apart from mosques.
- Protection from gender violence for women migrants seeking family reunification, whether they have official documentation or not.

## 3. Materials and Methods

### 3.1. Methodology and Participants

A prominent strand of intersectional research into Muslim women currently investigates identities as influenced by factors such as gender, religion, race, social class, ethnicity, etc., through which, depending on context, women can experience different situations as empowering and/or oppressive (Holley et al. 2016; Singh 2015; Biglia and Bonet i Martí 2017; Varley and Kaminski 2022). Here, however, we adopt a classical hermeneutic phenomenological approach (Folgueiras-Bertomeu et al. 2021) combined with a socially critical view articulated through the prism of transformation and applied empowerment (Zapata and Rondán 2016; Donato 2019). The first part of the study is based on the narratives elaborated on by the Muslim women of the Association, which were developed without the presence of or interference from any outside agents, in settings freely chosen by the women, around questions on their daily lives, devised by the group itself and based on very simple, minimally contextualized guidelines such as "the origin of your name", "important people for you", "your dreams", etc. The second participatory phase was framed in terms of the socially critical approach due to the practical tasks of organization and application and the resulting profound changes it caused in the participating women, the research team, the association, and the school.

The research project was conducted by means of a case study carried out among members and supporters of the Associació de Dones Musulmanes a Catalunya (Catalan Muslim Women's Association), a body legally registered in 2008. The group is based in Barcelona, defines itself as a meeting place for feminist Muslim women interested in learning about Islam, and brings together women of all nationalities and traditions. It represents a highly specific group of Catalan women: of Spanish nationality, mostly by birth but in some cases through long-term residence, with medium or high cultural and professional levels, professionally active and Muslim. In other words, the ADMAC mostly comprises women of Spanish nationality identifying themselves as activists with respect to their rights as citizens, feminists, and Muslims. The study population comprised the association's members and sympathizers who shared its goals and activities. Participants were involved not only as the objects but also as the subjects of research. Thus, the project put into practice an approach in which it was the Muslim women researchers themselves who gathered the data, made the initial analyses, formulated proposals, and participated in their joint evaluation with representatives of the other social actors, whose main responsibility was to assess the importance and relevance of the proposals emerging from the project. The role of the academic researchers was to train and guide the participants, and at the end of the project, to write the reports and summarize the policy proposals, which were then delivered to the Government of Catalonia in the form of a research report. Thus, the project, applying Responsible Research and Innovation (RRI) guidelines, combined three types of researchers (Table 5).

**Table 5.** Research team profiles and responsibilities.

| Groups | Research Team | Responsibilities |
| --- | --- | --- |
| ADMAC | 14 members of ADMAC | To interview 37 Muslim women association members (40%) and sympathizers (30%). The remaining 30% did not specify their relationship with the association. To participate in focus groups to formulate policy proposals. |
| GREDI | 6 senior researchers | Coordination, supervision, and consultancy |
| Social stakeholders | 17 political office holders, technical specialists, and academics | To analyse the importance and relevance of the proposals |

The case study was conducted with the participation of nine association members who volunteered to carry out both the individual and group activities. To this end, an intentional method of locating participants was applied, based on the motivation and availability of the Muslim women researchers. As they were already Spanish citizens with all corresponding rights, the participants aimed to leave aside the political and legal aspects (residence, political rights, etc.) and socio-economic factors (economic status, participation in institutions) of exclusion to focus on the cultural and religious dimensions. Thus, they fully represented this exclusion-prone religious minority.

Below we give details of some characteristics of the participants, who took the roles both of researchers and research subjects.

The women's origins were diverse: eleven were born in Catalonia, six in Morocco, two in the Gambia, nine in Spain, five in Latin American countries, and the remaining four in various Mediterranean countries. Their ages were diverse, with representatives of several generations, the most numerous groups being middle-aged, consistently with belonging to an association and playing an active role in the group under study (Figure 1).

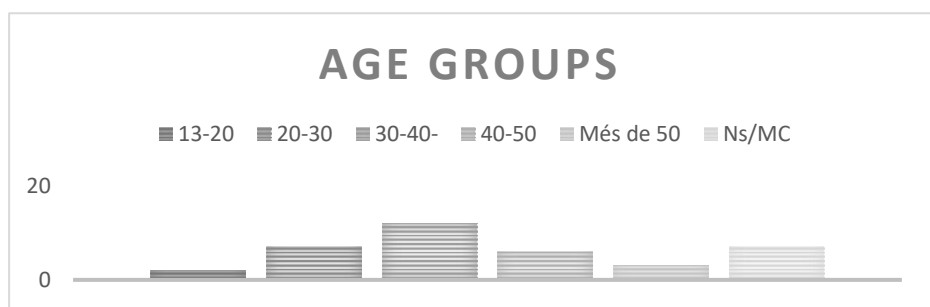

**Figure 1.** Participants' ages.

The marital status of the participants was also diverse. Sixteen percent were single, 43% married, and 3% widowed. It is worth noting the diversity of marital statuses and the presence of women "in a couple" (3%) or divorced (8%). Thus, in our view, the group adequately represents the reality of Catalan women's marital status.

There were also different situations in terms of children. Sixty-eight percent had children and 16% did not, while the remaining 16% did not specify. One woman had a daughter with a handicap, and another, born in Catalonia and converted to Islam, had two children adopted in Africa. The women with children had an average of 2.2 each. There was one woman with five children, six with none, and six others who did not specify.

Regarding the use of the hijab or veil by Muslim women, it is worth noting the diversity of postures towards this issue, so heavily charged with identity-related meanings (Figure 2).

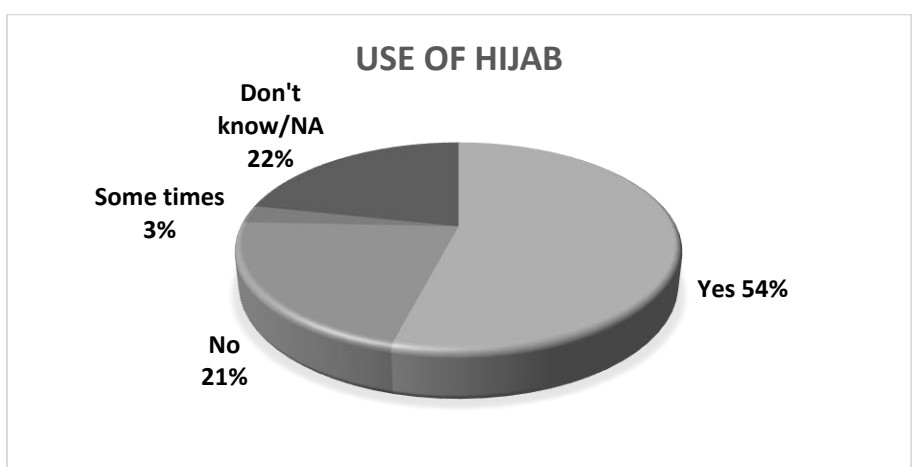

**Figure 2.** Participants' use of the hijab.

Regarding the employment status of the women, 60% worked with a contract (three civil servants and one self-employed professional), 16% were university students, 5.4% were unemployed, 2.7% were housewives, and 16.2% did not respond to this question. The sectors in which these women worked are shown below in Figure 3.

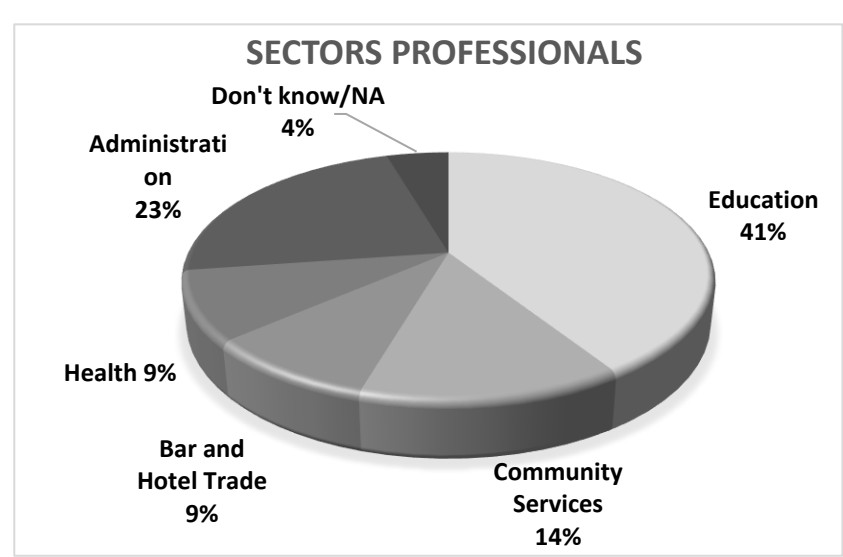

**Figure 3.** Participants' employment sectors.

Turning to participants' affiliation to Islam, 28% were born into the Muslim community, 35% had converted to Islam, and the other 35% did not specify.

Lastly, the languages in which ADMAC women expressed themselves were as follows: 43% Catalan, 49% Spanish, and 8% Arabic. More than half of the participants were residents of localities in the province of Barcelona, 40% of them residing in the city of Barcelona itself.

To accomplish the research objectives, the study was structured into two phases: (a) participatory research (PR), and (b) evaluation of the applicability to public policy of the proposals produced. The process of the project combined training and empirical work with concurrent communication and awareness-raising activities.

### 3.2. The Participatory Research Phase (PR)

This phase was carried out in two stages with different purposes. The aim of the first stage was to collect individual information using a biographical narrative approach of a phenomenological nature. The purpose was to make the Muslim women aware of their mental and conceptual frameworks and to have them reflect on these. The second stage

set out to compile and analyse in groups the emerging issues and the women's responses to them to gain a comprehensive overview of their social analyses. The plurality and complexity of their views, identities, and positions as citizens within the framework of common values were appreciable. The main outcome of this analysis was the ranked summary of proposals put forward by the ADMAC. Two instruments were used in this phase: the first was the existential interview devised by Weiss et al. (2015), a questionnaire comprising open-ended questions on women's everyday spiritual life, and the second was discussion groups organized around the project's main topic areas. These groups were made up of the core group of Muslim women researchers and a representative number of the women who had been interviewed in the first phase. The discussion groups (organized first as a large group and then according to the different topic areas) held several sessions to analyse, prioritize, and summarize the issues and proposals emerging from the content of the interviews. The analytical procedure of this stage involved a series of activities aimed at organizing their words and phrases (both deductive and inductive); sharing, validating, and enriching the syntheses; ranking the proposals; and, finally, drawing up the summary in the form of specific proposals for the topic areas under study.

*3.3. Phase Assessing the Applicability of the Proposals to Public Policy*

This phase was also carried out in two stages. First, the policy proposals emerging from the participatory research phase were analysed according to the Ivàlua logical framework criteria (Cordoncillo 2022). Ivàlua is a consulting firm specializing in the assessment of public policy, put forward for the study by the Catalan School of Public Administration (EAPC). Its specialists trained both the project's academics and the Muslim women researchers of the ADMAC, advising them on how to ensure that the proposals matched the criteria.

The proposals were prescriptive and responded to the evaluation criteria (specific problems, quantifiable indicators, premises, limiting factors, etc.). The analysis was performed by the team of Muslim women and academic researchers over several joint sessions. The second stage of the applicability assessment was carried out via three roundtable discussions in which ADMAC representatives, academics, professional specialists, policymakers, and leading citizens analysed and validated the proposals according to criteria of importance, relevance, and impact.

The information emerging was both textual (memos summarizing the existential interviews and the focus groups) and audio–visual (audio recordings of the interviews and video recordings of all the group sessions). The textual information was analysed manually by the Muslim women researchers and in other cases by the university research team using Nvivo10 software. The method of analysis was qualitative. An inductive analysis was used in the participatory research, in which the major categories emerged from two sources: firstly, the research questions and premises themselves, and secondly the interests raised by the Muslim women researchers. The deductive criteria of the Ivàlua logical framework (Ivàlua 2020) were applied in the evaluation.

**4. Conclusions and Discussion**

This article presents the process and results of a study that was not only about Muslim women but also conducted among and by Muslim women, a study in which a group of Catalan Islamic women, feminists, and activists was fully recognized and given voice, and in which the research team worked horizontally. Here, we have described the participatory process and its outcomes. The results are framed as political proposals intended as guidelines for the Catalan government's Immigration and Citizenship Plan, since this was the purpose of the research; i.e., to generate precepts and proposals for fostering a multicultural identity in Catalonia.

The project's objectives were (1) to gather the real experiences of Catalan Muslim women with respect to their social, family, and spiritual lives; (2) to analyse and synthesize dialogically the challenges faced and responses to them emerging from this experiential

view of the women of the Catalan Muslim Women's Association (ADMAC) in the areas of services and policies; and (3) to formulate specific policy proposals in the fields of education, religion, and gender.

In assessing the project methodology, we would like to highlight a number of points here. Regarding the group selected to be both the object and subject of study, the choice of the ADMAC effectively addressed the express intention of not linking their barriers to acceptance and integration to socio-economic and political–legal factors. The association, which brings together empowered, qualified women, many of them converts, was able to overcome certain intersectional biases and obtain alternative, more critical views, situated at the core of Catalan society. In addition, by giving the role of interviewers to the women themselves, we set out to establish a power relationship that would be different not only from that typical to research interviews conducted by university researchers, but also to those in society itself. We hypothesized that this horizontality could overcome acquiescent response bias and social prestige, amongst others, and would allow narratives with a slightly different content to emerge. Among the women, there were no images to protect, no group agenda to promote, no expectations to satisfy, no postures to justify.

The process of the development and implementation of the participatory research made the intentions underpinning the initial design explicit, and this had an impact not only on the resulting content but also on the methodological approach:

1.  Gathering experiential, meaningful information as horizontally as possible about what being a woman, a Muslim, and a member of a Muslim community in Catalonia means and involves.
2.  Identifying, in a divergent way, through endeavouring to overcome acquiescent response bias, social prestige, and power and status differences, the concerns, problems, responses, and alternatives expressed by women from the standpoint of their own experience, in the fields of social services, gender, and religious and spiritual education.

This process was extremely intense, complex, and full of moments not only of fulfilment, but also of insecurity and tension. We would like to highlight here some of the views expressed by the women Muslim researchers in the assessment and closing sessions:

*   "Freedom and courage to dare to say what we think, feel, and believe";
*   "Reality is not linear";
*   "They're issues that are scary to address openly and that's why, in the end, they're not touched on, they're hidden and concealed";
*   "It's a very complex and delicate issue, which touches on a lot of nerves, where there are a lot of contradictions in play for all actors";
*   "There's a structurally unjust framework (for example, regularization of women for family reasons that makes them absolutely dependent on their husbands) and not egalitarian (the false non-confessionalism of the state)".

In relation to the issues raised in the study, it is worth mentioning the phenomenological approach, which focuses on the participating women's experience without taking into account identity variables such as age, education, origin, etc. Although religious issues were not explicitly addressed, opinions, practices, and emotions stemming from the participants' experiences as Muslim women arose spontaneously. At this point, it is crucial to remember the major importance of spirituality for almost all of these women, at the same time as their reticence, dissatisfactions, and concerns regarding their own community religious practices. Thus, issues other than the attitudes of non-Muslim society were addressed, with participants expressing complaints about the majority Muslim community and desires for change in its model, especially in relation to their children's education and the satisfaction of their spiritual needs.

Regarding the activities undertaken, we wish to stress the creativity and the social and community impact of the resulting proposals, which ranged from the creation of new

professional profiles, capable of satisfying new social needs, to changes in the curriculum and education system, even new social structures.

Regarding the changes expected with the implementation of the proposed religious, education, and gender policies, it seems to us especially significant that the entire process of dialogue between various interlocutors (the women in the group, specialists, academics, and regional government officers), conducted over months, with simultaneous training in public policy, enabled the participant–researchers to transcend their own problems, and their political agency as an association, and to put forward systemic and structural changes that would affect the whole of Catalan society with a view to enable greater recognition and inclusion for all identities.

In the light of these results and the lessons learned during the project, it is worth underscoring, as we hope to have demonstrated, that citizen feminism was put into practice throughout. In seeking to analyse the extent to which women in different communities exercise their citizenship in a broad sense, feminist citizenship studies have shifted the term away from its former narrow legal–political basis, redefining it as a wider and more inclusive socio-cultural concept. As Predelli et al. remembered "The classical view of citizenship as the delineation of political-legal rights and duties has been challenged on several fronts, including its limitation of citizenship in the public sphere and its narrow view of citizenship as a "status-situation (2019, p. 54)". These same authors remembered that feminist scholarship has argued that citizenship includes practices in all spheres of life (political, economic, social, cultural, religious, domestic, and personal). A radical implication of feminist theories of citizenship is that our practices should be inclusive of women and men in all aspects of life.

This study has attended to the voices of a group of women identifying themselves as both feminists and Muslims. They are believers, activists in disseminating the idea that Islam promotes gender equality, in line with Nurmila (2011) and Scott (2009). They put forward critical, empowering proposals with regard not only to the discriminatory behaviours and situations that they perceived in their daily lives and in relationships with the Catalan state, but also to how their own Muslim communities approached women and their spirituality. The ADMAC members' views, analyses, and proposals concur with the analysis of Navarro-Granados (2022): that the basis of discrimination against Muslim women stems fundamentally from the patriarchal culture and traditions prevailing in most Muslim-majority countries (Alhareth et al. 2015) and from interpretations of the Qur'an made mainly by men (Lussier and Fish 2016). The practical reconciliation of these issues is complex; there are no hegemonic positions, as we find in general in other spheres of feminism, religion, and society. The project participants recognized themselves as occupying the position described by Mendoza (2018); i.e., they rejected literalist interpretations of the Qur'an and argued for the historical, political, and social contextualization of its scriptures. In the specific group studied, religious identities, like others, were not unitary, stable, or fixed. Identity is an ongoing project that we constantly work on, reshaping and recasting it in the interactions between our lives, our personal agency, and the social and institutional contexts in which we are situated. In Linda Alcoff's words, "Social identities are not imposed from the outside, but are best understood as sites from which we perceive, act, and engage with others. These sites are not simply social locations or positions, but also hermeneutic horizons shaped by experiences, core beliefs, and community values that influence our orientation and responses to future experiences" (Alcoff 2006, p. 5). Identities, including religious and gender identities, are constructed through a dynamic interplay between internal (within the individual self) and external processes, which cannot easily be observed or separated. We find the notion of a "plurality-integrating identity" useful, as it captures the floating (divergent) and plural aspects of identity in addition to its stable and connected (more deeply assimilated) aspects. According to Østberg (2003), the identities can be plural and can be expressed depending on the specific requirement of the context (situational identity). Through the existence of such fragmented and diverse identities, all

of us build, integrate, and recreate these diverse pieces assuming the contradictions that can be identified by us and others.

To conclude, and by way of a discussion, the idea that religion and citizenship are connected is not new as highlighted by Predelli (2008). The term "religious citizenship" seems to be increasingly taken up by scholars; for example, Hudson (2003, p. 426) distinguishes between definitions of religious citizenship by the nation state and civil society, and based on the "rights of individuals". In this area, we follow the approach of Lister (1997) about religious citizenship encompassing three dimensions: (a) status and rights; (b) participation; and (c) identity and belonging. In conclusion, religious identities can be agents as barriers to citizenship, depending on the references adopted by people. Religion is therefore a flexible resource that can shape identity, with both empowering and disempowering effects.

This study, conducted among the members of the Associació de Dones Musulmanes a Catalunya (Catalan Muslim Women's Association, ADMAC), has provided an alternative perspective from which to understand these processes, which are increasingly diversified and multidimensional. Through the study, we were able to trace the spiritual, political, and civic dimensions of a group of Muslim women who have empowered themselves, defended their political agency as an association, and been able to contribute their own political proposals, in which education plays a major role. Whether as a sphere of debate and action, as a terrain for recognizing rights, or as a means of exercising citizenship, the area of Muslim women's religious and civic rights deserves the full attention of academia, political organizations, and bodies responsible for interreligious dialogue, responding to the challenge of gender inclusion highlighted by UNESCO (2018).

**Author Contributions:** Conceptualization, A.A., M.S.S. and N.L.R.; methodology, A.A., M.S.S. and A.A.P.; software, A.A.; validation, A.A., M.S.S., A.A.P. and N.L.R.; formal analysis and investigation, A.A., M.S.S., A.A.P. and N.L.R.; writing—original draft preparation, A.A. and M.S.S.; writing—review and editing, A.A.P. and N.L.R. All authors have read and agreed to the published version of the manuscript.

**Funding:** This study was funded by the EAPC "Research Engine" programme, set up by the Escola d'Administració Pública de Catalunya (Catalan School of Public Administration, EAPC), with the number 6030-2019-s367.

**Institutional Review Board Statement:** The Institutional Review Board of the Research Area of Catalan School of Public Administration (ECEAPC) whose principal was N. Guevara signed the protocol code GGE760024/2019 in 2 April 2019.

**Informed Consent Statement:** Participants' signed consent was obtained before the beginning of the project, including image rights, confidentiality in the research process, and cessation of the resulting information to the Catalan government.

**Data Availability Statement:** The school retains ownership and stockage of the information and data.

**Acknowledgments:** We wish to acknowledge the commitment and motivation of the Catalan Muslim women of the ADMAC and all the specialists who participated in the study.

**Conflicts of Interest:** The authors declare no conflicts of interest.

## Note

[1] On 17–18 August 2017, there were two terrorist attacks claimed by the Islamic state (IS) organization, one on the Ramblas in Barcelona and the other in Cambrils (Tarragona). In the capital of Catalonia, 16 people were killed and more than 95 were injured; in Cambrils, six people were injured and one terrorist was killed.

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
