# Peer review of "Speak Catalan to Me, I’m a Catalan Muslim Woman: Producing Proposals for Religious and Education Policy through Participatory Research from a Gender Perspective"

_religions, doi:10.3390/rel15020141_

Round 1

Reviewer 1 Report

Comments and Suggestions for Authors

The chosen theme seems to me to be very interesting and necessary. 

- Should update the article, with 2023 citations.

- I believe that the article could be improved by introducing more discussion on what policies or initiatives should be carried out to achieve better inclusion of these minorities in Catalonia.

- In the theoretical framework, I would add research on the theory of intersectionality, which tells us how racialised Muslim women experience different types of discrimination at the same time. 

- I am concerned about data protection in the photographs shown, I think it is not necessary and violates participants' anonymity

- I believe that the number of references should be considerably increased for a quality scientific article. 

I recommend the following research on intersectionality and Islamic feminism: 

- https://journals.eagora.org/revHUMAN/article/view/4126

- https://doi.org/10.1080/01596306.2019.1709157

Author Response

Dear Reviewer

Thank you for your review, it has helped us to improve very munch our paper.

Reviewer 2 Report

Comments and Suggestions for Authors

In the article, the authors provide an exemplary transparent account of the methodology employed and the study's objectives. Specifically, the goal is to contribute a policy proposal to the Catalan government on how Muslim women can participate in Catalan/Spanish society. The study adopts a community-based approach, with participants, in this case, members of the Catalan Muslim Women’s Associations (ADMAC), actively conducting the research, while the responsible researchers assume the role of partners. The authors argue that this approach fosters a more horizontal relationship with informants compared to the traditionally vertical role researchers have with their subjects.

The study is framed within a postcolonial worldview, assuming that Muslims, especially Muslim women, face extensive discrimination. This perspective has become a truth in much of contemporary social science research on ethnic and religious minorities. However, the study lacks data on the extent and specific manifestations of such discrimination. The presented data on women's positions in the Catalan workforce, despite many wearing hijabs, contradicts the prevailing notion, suggesting that these women have successfully integrated into the job market. It would be pertinent to clarify, given that discrimination is a driving force for the study, whether the women's issues with discrimination are solely connected to the established majority society or also pertain to their lives within the Muslim community.

Despite addressing a religious group, Muslims, little is said about religion in the text. While religion is mentioned, readers are not provided with specific details about the religious interpretations the women hold or the role of religion in their lives. Generally, there are religious interpretations among Muslims that may hinder full participation in European societies, such as the view that women and men should be kept separate or restrictions on friendships and marriages. A higher degree of theoretical problematization concerning concepts like integration, participation, and Muslim feminism would have been desirable. However, it may be inherent in participatory research that extensively problematizing various aspects is challenging, given that such research relies on researchers assuming the role of an "allied other" to the participants.

The article has several merits, as initially acknowledged. However, my assessment is that the authors adopt a perspective that is overly unproblematic on issues related to Muslim participation in the established society. The content of the article is also not particularly original, as much of the knowledge presented has been previously discussed in a continuous stream of scholarly and popular science texts. Additionally, the article provides a rather superficial depiction of religion, making it less suitable for publication in a journal like "Religions." Nevertheless, in terms of methodological transparency and clarity of objectives, the article sets an admirable example.

Comments on the Quality of English Language

No comments

Author Response

Dear reviewer

Thank you for your comments. All of them have helped to us in the improvement of our paper.

Reviewer 3 Report

Comments and Suggestions for Authors

I wish you the best in future research. 

Author Response

Dear reviewer, thank you for your positive and kind review.

Round 2

Reviewer 2 Report

Comments and Suggestions for Authors

I have no further suggestions of improvement. Still, I think that the article  is in need of more religious content, however, at the same time you have written a methodologically transparent case-study from a Catalan context which contributes to established scholarship.

Comments on the Quality of English Language

I have no comments on language.